# Emerging Trends in Porogens toward Material Fabrication: Recent Progresses and Challenges

**DOI:** 10.3390/polym14235209

**Published:** 2022-11-30

**Authors:** D. Shanthana Lakshmi, Radha K. S., Roberto Castro-Muñoz, Marek Tańczyk

**Affiliations:** 1Ariviya Deep Tech Private Ltd., #202, TBI, PMIST, Thanjavur 613403, Tamil Nadu, India; 2Division Head, Chemistry, R.M.D. Engineering College, Chennai 601206, Tamil Nadu, India; 3Department of Sanitary Engineering, Faculty of Civil and Environmental Engineering, Gdansk University of Technology, G. Narutowicza St. 11/12, 80-233 Gdansk, Poland; 4Tecnologico de Monterrey Campus Toluca, Av. Eduardo Monroy Cárdenas 2000 San Antonio Buenavista, Toluca de Lerdo 50110, Mexico; 5Institute of Chemical Engineering, Polish Academy of Sciences, Ul. Bałtycka 5, 44-100 Gliwice, Poland

**Keywords:** porous material, porogens, polymers, ceramic, biopolymers

## Abstract

Fabrication of tailor-made materials requires meticulous planning, use of technical equipments, major components and suitable additives that influence the end application. Most of the processes of separation/transport/adsorption have environmental applications that demands a material to be with measurable porous nature, stability (mechanical, thermal) and morphology. Researchers say that a vital role is played by porogens in this regard. Porogens (i.e., synthetic, natural, mixed) and their qualitative and quantitative influence on the substrate material (polymers (bio, synthetic), ceramic, metals, etc.) and their fabrication processes are summarized. In most cases, porogens critically influence the morphology, performance, surface and cross-section, which are directly linked to material efficiency, stability, reusability potential and its applications. However, currently there are no review articles exclusively focused on the porogen pores’ role in material fabrication in general. Accordingly, this article comprises a review of the literature on various types of porogens, their efficiency in different host materials (organic, inorganic, etc.), pore size distribution (macro, micro and nano), their advantages and limitations, to a certain extent, and their critical applications. These include separation, transport of pollutants, stability improvement and much more. The progress made and the remaining challenges in porogens’ role in the material fabrication process need to be summarized for researcher’s attention.

## 1. Introduction

Nowadays, porous materials are gaining significant interest in various fields due to their wide range of applications in all fields such as filtration membranes, catalyst supports, adsorbents, thermal energy, insulation systems, evaporation systems, biomaterials scaffolds for bone ingrowth and drug delivery. In general, whatever the application, there is always a compromise between porosity and mechanical strength of the final material. Hence, the range of porosity, pore morphology and pore size distribution are to be optimized for specific applications. This is controlled and determined based on the fabrication method selected. In material designing, the fabrication process requires stringent regulations and a meticulous planning process, so that the end product can fulfil a wide range of applications, i.e., stability, tailor-made structure, reusability, cost-effectiveness, etc. For the past few decades, there has been enormous interest in the application of porous materials ranging from separation science to medicine, and its synthesis has roots in interesting theory and experimental facts. The morphology of a material and the topology of pores are entire research areas. Porous materials created by nature or by synthetic design have found great utility in all aspects of human activity [1].

The history and importance of porous materials are long and varied, hailing from ancient times, and one such material is porous charcoal, which has applications in the medicinal field. An ever-increasing awareness in civil society and environmental regulations of global bodies have revived the topic of porous materials, which have a versatile range of applications, as mentioned earlier.

Classification of pores is one of the basic requisites of comprehensive characterization of porous materials such as catalysts, adsorbents, oxides, carbons, zeolites, organic polymers, soils, etc., based on their structure, size, accessibility and shape parameters. Pores are classified as per IUPAC regulations: micropore (<2 nm), mesopore (2–50 nm) and macropore (>50 nm) [2]. Material pore size and its distribution are crucial parameters for the selection of application domains, such as separation, water purification, catalytic support, energy storage and capacitors. A specific application requires material with an appropriate pore type. They are highly dependent on the external/internal pore structure, so it is wise to decode the morphology and structural arrangement of porous materials to understand the physical processes involved; these can be evaluated through internal geometry, size, connectivity, etc.

The design and synthesis of organic, inorganic and polymeric materials with controlled pore structures are important in both academic and industrial sectors due to the ever-increasing demand in potential industrial applications. Therefore, it is the purpose of this review to provide an introductory description of pores, porous material types and the role of pore forming agents (porogens) in various industrial and academic applications. In connection with this, different materials (with different physical and chemical natures) which can act as a porogen on various host surfaces, i.e., polymer, ceramic, glass, metals, etc., are summarized, especially with respect to the enhancement of pores and pore size and their direct impact on applications. However, currently there are no review articles specifically focused on porous media fabrication based on various porogens. The objective of this work is also to review the types and actions of porogens in developing porous materials, and the application of porogens. In this review, we summarize recent reports on the types of porogens (organic, inorganic, ceramic, etc.), nature of porogens (synthetic, bio/natural) and the type of host (polymers (synthetic, bio), glass, ceramic, metal, etc.) substrate material which requires porogens.

## 2. All about Pores

Porous materials contain pores, also called voids, either in isolation or interconnected, that may have similar or different shapes and sizes to form complex networks of channels, which are usually filled with fluid under normal atmospheric conditions, such as air, liquid water or water vapor. The pore structure is developed during the crystallization stage of the solid, or by subsequent treatment. According to Kaneko classification, pores are classified based on the following geometrical shapes (Figure 1): cylindrical, slit-shaped, cone-shaped and ink bottle-shaped pores. Rhomboid, elliptical and square are the other pore shapes reported in the literature. Pore shapes fall under different geometric bodies such as cylinders (for activated oxides such as alumina or magnesia), prisms (fibrous zeolites), cavities and windows (other zeolites), slits (clays and activated carbons), or spheres (silica gel, zirconia gel, etc.) [1,2,3].

Pore size is defined as the pore diameter or width (distance between the two walls) which has an accurate meaning only with a defined geometrical shape and analyzed through gas sorption isotherms. According to the IUPAC definition, porous materials are classified into three major categories depending on their pore size, as follows:Microporous materials (diameters up to 2.0 nm);Mesoporous materials (diameters between 2.0 and 50.0 nm);Macro porous materials (sizes exceeding 50.0 nm) [1].

It may be desirable to subdivide micropores into those smaller than about 0.7 nm, as narrow micropores or ultra-micropores, and those in the range from 0.7 to 2.0 nm, termed as supermicropores. Material porosity is a measure of the void spaces in a material, and is a fraction of the volume of voids over the total volume, with a value between 0 and 1. In the case of biomaterials loaded with drugs, controlled porosity allows their liberation in a targeted site by a slow, continuous and controlled flux over a certain period. Such a controlled drug delivery is linked to microporosity, which are materials consisting of pores with a size less than 10 mm in diameter. The field of bioceramics demands materials with an additional larger porosity (macroporosity) for promoting integration with biological tissues [4]. Carbon-based materials, used as adsorbents, may reveal uniform or non-uniform pore size distribution, with pore diameters ranging from micro- to macro-pores. Commercial zeolites, used as molecular sieves, are in the category of ultra-micro pore size, which is 0.3–0.5.nm. In contrast, silica gels having larger pores or macropores up to 300 nm are also known.

### 2.1. Porous Materials

Porous materials, either natural or artificial, have attracted the attention of many researchers owing to their potential applications based on their unique properties such as high surface area, relatively low stiffness, shape selectivity, permeability, etc. Based on their composition, these porous materials can be classified into two types: inorganic and carbon-based materials. In the past two decades, molecular design has been increasingly prevalent in porous materials such as zeolites, metal–organic frameworks (MOFs), covalent organic frameworks (COFs) and porous polymers (membranes) which have widespread applications (Figure 2) in adsorption, catalysis, separation, purification and energy storage and production [5].

Porous materials have created a significant contribution to society and they are still developing rapidly. They must be scalable and should satisfy the multiple functional criteria such as stability, selectivity, adsorption kinetics, reusability and processability, mechanical properties and thermal properties, while keeping costs low. Hence, there is a great challenge in the process of designing; to have control over pore structure and also to understand the structure–property relationship in a detailed manner. Design concepts in porous materials have advanced rapidly in recent years as a result of the latest developments in materials characterization, modular synthesis and computational structure–property predictions. As we are all aware, there is no one-size-fits-all solution; the real challenge exists in choosing the right type of porous material for a given application. In addition, porous materials are not only used, but also have potential, in electronics, light harvesting and energy, proton conduction, molecular sensing applications, etc. Hence, there is a lot of scope to explore new porous materials.

### 2.2. Action of Porogens 

The primary task in the fabrication of porous materials is to achieve a controllable or tailored pore size and porosity using different methodologies, i.e., physical techniques, chemical agents/modifications, etc. [6]. The addition of chemical compounds as additives during material fabrication is a viable option in the development of porous materials, which is crucial for a plethora of applications. The terminology “porogens” and “pore formers” refers to a material additive that has the ability to disperse in the feed composition and may leach out after the fabrication stage. In some cases, pore formers develop pores during fabrication, but the concurrent leaching may alter the stability and efficiency of the materials. Thus, by controlling the properties of the porogens, the microstructures of the host materials can also be tailored, which is critical for the fabrication of porous materials. Therefore, a systematic study was needed to understand the relations between the properties of the porogens, the resultant pore structures and the dimensional changes upon their removal. It was also demonstrated that the thermal and morphological properties of the porogens have a vital effect on the features of the resultant microstructures [7]. In the field of membrane technology, porogens play a very vital role as a hydrophilic additive, which increases the separation performance of the membrane which in turn may be useful for controlling fouling properties.

The bar diagram (Figure 3) shows the significant research articles published on pore-forming materials and porogens in recent years.

#### Types of Porogens

Porogens are generally used to prepare the porous support, verified by theory and practice, and the major division is organic and inorganic. Examples of inorganic porogens are ammonium carbonate, calcium carbonate, ammonium bicarbonate and ammonium chloride, etc. (Table 1), while examples of organic porogens are sawdust, shell powder, starch, polystyrene, water-soluble polymers such as Poly ethylene glycol (PEG), Polyvinyl pyrrolidone (PVP), Polyvinylalcohol (PVA), Polymethaacrylate (PMA), Polyacrylicacid (PAA), etc. Nowadays, bio/green porogens are attracting the attention of researchers. Activated carbon from various sources, moringa seed powder and marine-derived polymers such as chitosan, κ- carrageenan, alginate and ulvan have been employed as porogens in recent studies. In addition, green seaweed-derived sulphated polysaccharides have proven to be superior porogens compared to the others, especially when used in minimum quantities.

## 3. Inorganic Porogens 

Polymeric membranes suffer from a relationship trade-off between selectivity and permeability. In order to overcome this trade-off, different approaches have been applied by changing the membrane properties such as pore size, porosity, hydrophilicity, surface properties and polymer morphology. Inorganic fillers and additives have been extensively used for tuning the membrane properties by incorporating them into polymer doped solutions. Several inorganic nanomaterials such as SiO_2_, TiO_2_, Al_2_O_3_, Fe_3_O_4_, CaCO_3_, graphene oxide, carbon nanotubes, zeolites and CeO_2_ have been utilized to tune the membrane properties (Table 2).

Similar observations were noticed in almost all the research investigations when incorporating inorganic nanoparticles into thin film composite membranes with respect to water flux and solute flux. In almost all thin film nano composite membranes investigated so far, regardless of the fabrication method, the incorporation of inorganic nanoparticles (SiO_2,_ functionalized MWCNTs and TiO_2_) resulted in very high forward osmosis performance, leading to an increase in the roughness and hydrophilicity of the TFN membranes along with the best salt rejection and a high water flux [9,11,12]. One such measurement and observation is shown in Figure 4. A concentrative internal concentration polarization (ICP) occurred when the active rejection layer faced the draw solution (AL-DS orientation), resulting in the rejected feed solute accumulating in the support layer. Similarly, a dilutive ICP occurred due to dilution of the draw solution inside the support layer when the active layer is placed against the feed solution (AL-FS orientation). This phenomenon reduced the effective driving force. Since it occurs inside the support layer, it could not be eliminated by increasing the flow rate turbulence. Therefore, in order to minimize ICP, a small structure parameter is preferred for the support layer to fabricate an appropriate FO membrane [9]. Similarly, in the study for the potential use of mixed matrix membranes (MMMs), a substantially improved substrate porosity, mass transfer coefficient, as well as rejection layer properties, were observed. This gave a new and additional dimension for ICP control in osmotically driven membrane processes [10].

Furthermore, to add more support to this finding, a zeolite loaded TFN membrane demonstrated significantly enhanced FO water flux due to their improved water permeability, and it was proved that this is potentially more favorable during the application of treating feed solutions with water with relatively high salinity under AL-FS orientation [14].

However, the inorganic nanoparticles are liable to aggregate in the membranes at higher loadings/concentration due to the difference in material compatibility with polymeric chains [15]. Agglomeration of the nanofillers could develop defects and voids, which leads to the free passage of penetrants with the reduced separation efficiency of the membrane. The addition of a two-dimensional smectite-type clay and one-dimensional nanowires was also used to avoid the agglomeration and leakage of nanoparticles [16]. The addition of nanowires instead of nanoparticles in the casting solutions revealed many varied results in the UF membrane performance, as highlighted below in Figure 5.

Monticelli et al. introduced different types of clay in polysulfone (PSf) membranes and observed that Cloisite Na and Cloisite 93A formed microaggregates which affected the phase inversion process in the coagulation bath. They also found that it enhanced the wettability and mechanical properties of dense films [17]. 

Inorganic salts, especially lyotropic salts of lithium, zinc, calcium and magnesium with bromide, iodide, nitrate, thiocyanate and perchlorate—such as lithium chloride (LiCl) and lithium perchlorate (LiClO_4_)—are known for generating porosity, good interconnectivity and increasing the coagulation rate of polymer-doped solution [18,19,20]. Among different types of lithium halides, lithium bromide (LiBr, LiCl) and lithium fluoride (LiF) are used as additives. The addition of LiCl and LiF tends to enhance the viscosity of the casting solution, which suppresses the macro-void formation with excellent interconnectivity and porosity along with high fluxes and higher rejection rates. However, the addition of LiBr in polyethersulfone (PES) polymer-doped solution resulted in moderate diffusion of the solvent during the coagulation process and delayed the solvent–nonsolvent demixing phenomena. It mainly affected the membrane surface layer rather than the microstructure of the membrane; LiBr acted as a pore inhibitor rather than a pore former [21].

In general, the concentration of salt additives cannot exceed a certain limit due to solubility limitations. However, the concentration of salts can be increased by the addition of different salts, preferably one that has no common ion with the first salt. Such a combination of two salts, such as ZnCl_2_/Pyridine Hydrochloride and Mg(ClO_4_)_2_/Pyridine Hydrochloride, significantly improves the flux of the membrane (Table 3). In addition, higher additive concentrations resulted in a more open structure, with the asymmetric arrangement clearly established from the investigations [22].

In further investigations with a wider range of inorganic materials incorporated in nanofiltration or loose nanofiltration membranes, a significant increase in the pure water flux and hydrophilicity of the membranes was observed. Membranes embedded with a 0.5 wt.% novel metformin/GO/Fe_3_O_4_ hybrid (MMGO) acted as the best membrane compared to bare PES membrane for water permeability enhancement and removal of copper ions and dye [24]. In another study, Chitosan–Montmorillonite nanosheets blended with NF membranes exhibited remarkable antifouling properties, demonstrating CS–MMT nanosheets as an excellent antifouling material as well as having very good mechanical stability. This demonstrates that CS–MMT loose NF membranes are an ideal choice for dye purification under low pressure and with high efficiency [25]. Similar observations were obtained with very small percentage addition of inorganic porogens such as sulfonated halloysite nanotubes, CNTs, etc., which showed enhanced hydrophilicity and charge density. They exhibited the highest flux coupled with high rejection of salts and reactive dyes and low retention of the saline solution [26,27].

In a study related to UF membranes, the combination of organic–inorganic porogen PEG 600/LiCl caused the resultant membranes to exhibit an increased average pore radius and surface porosity compared to the CA/SiO_2_ membrane without additives. The presence of PEG 600 and LiCl in the doped solution improved the permeate flux of proteins. With PEG 600 as the porogen, the reversible fouling resistance ratio decreased to a greater extent. In addition, PEG 600 proved to be the better porogen in comparison to LiCl, by offering more resistance to total fouling thereby increasing the flux recovery ratio and recycling potential of the CA/SiO_2_ blended membranes [28]. In one more investigation of flat sheet PSf/clay nanocomposite membranes using clay as a porogen, better mechanical properties, high stability, improved hydrophilicity of membranes and an increased ratio of large pore in the skin layer was demonstrated [29].

## 4. Organic/Polymeric Porogens

There are a wide variety of organic porogens used in various fields of research. First, let us discuss the synthetic organic polymers such as Poly ethylene glycol (PEG), Polyvinyl pyrrolidone (PVP), Polyvinylalcohol (PVA), Polyacrylamide (PAM), Polyacrylic acid (PAA) and N-(2-Hydroxypropyl) meth acrylamide (HPMA), which play a vital role in pore formation, distribution of pores and size of the porous materials and more specifically in the fabrication of membrane materials (Table 4). The type and amount of porogen added have a critical impact on the porosity of polymeric membranes, offering wide applications.

### 4.1. Organic Water Soluble Porogens

#### Water-Soluble Porogens in Polymeric Microspheres (MCs)

Polymeric microspheres are small spherical micro particles, with a wide variety of properties such as bulk total interfacial area, large inner volume, spherical shape and better stability suitable for chemical loading [54]. They are used in significant applications such as controlled release of encapsulated drugs, masking of odor and/or taste of encapsulating materials and isolation of encapsulating materials. The porous surface of the polymeric MCs depends on the nature of the solvent and the effect of the pore former and its concentration.

Two different synthetic polymers, polyether sulfone and polyetheretherketone with card (PEEKWC) MCs were fabricated with and without pore formers using four different solvents: N,N-Dimethylformamide (DMF), Dimethyl sulfoxide (DMSO), N-Methyl-2-pyrrolidone (NMP) and γ-Butyrolactone (GBL). The asymmetric, symmetric, porous, spongy and finger-like structure of the MCs varied based on several physical and chemical properties. It was possible to design and synthesize a polymer with controlled size, smoothness, etc., with structural guarantee [55]. The optical image is MCs formed is shown in Figure 6.

The porous structure of the MC increased with an increase in PVP concentration. The polymeric microspheres exhibited a central cavity and an asymmetric (finger type) structure having both a porous as well as a dense skin layer at the shell side (Figure 7).

### 4.2. Ionic Liquids as Porogens

In a similar investigation, ionic liquids (IL) were explored for the pore formation process in polymer membranes and microspheres using the phase inversion technique. Any tailor-made material fabrication process requires meticulous planning where each component fulfils an active role in determining the size, morphology, stability and application. A membrane or microsphere (MC) that is composed of ILs and polymers as core components without physical/chemical interaction generates the final products. In the case of ionic liquids, physico-chemical properties (viscosity, density, etc.), nature (carbon chain length, cations and anions), toxicity, cost and availability are considered as crucial factors. The choice of the materials (membrane, microsphere) was based on destined applications, such as separation, adsorption (organic, inorganic compounds), target delivery, loading of different chemicals, etc. 

However, the novel composite materials using IL incorporated MCs as porogens showed a higher uniformity compared to that of microspheres without IL. These may create varied opportunities for the preparation of extraction capsules and microreactors with a smaller size and with a mean diameter value of 1.1 mm (Figure 8) [55].

Another venture was attempted by the same researchers for the fabrication of membranes and microcapsules using hydrophilic porous polyethersulfone (PES) with 1-butyl-3-methylimidazolium hexafluorophosphate ([BMIM][PF6]) as the structure control agent (porogen), instead of the conventional PVPK17. The addition of 2 wt.% PVPK17 increased the pore size, more specifically, the hydrophilicity and porosity. In the case of ionic liquid membranes (ILMs), 5–25 wt.% [BMIM][PF6] was incorporated in the PES-doped solution, whereas for ionic liquid microcapsules (ILMC), the concentration of [BMIM][PF6] was restricted to 5–15 wt.% due to the IL viscosity range. [BMIM][PF6] concentration was a crucial parameter in deciding the porosity and morphology; moreover, a higher concentration of [BMIM][PF6] enhanced the viscosity thereby reducing solvent exchange speed and channeled structures (finger-like) to a spongy morphology (Figure 9). Hence, these novel composite materials with ILs displayed excellent solubility for a broad range of organic molecules and they were used in the immobilization of suitable extractants into microcapsules and membranes [56].

Yet another study was carried out by the same authors using ionic liquids in polymer inclusion membranes (PIMs) based on polyether sulfone (PES)/1-butyl-3-methylimidazolium hexafluorophosphate ([BMIM][PF_6_]) using non-solvent induced phase separation (NIPS), tested for the removal of reactive blue 19 (RB_19_) as a model anionic dye. Dye adsorption efficiency was simulated by statistical analysis, which depended on pH, contact time, initial dye concentration and the amount and weight of adsorbent (PIMs). As in the previous investigation by the authors, the addition of 2 wt.% of PVPK17 increased the pore size to a greater extent, but the addition of [BMIM][PF_6_] (less than 15 wt.%) did not influence membrane properties (Figure 10 and Figure 11). On the contrary, when the concertation of IL (25 wt.%) increased, the morphology of the PES membrane drastically changed from finger to spongy type. Thus, the PIMs made of PES/2 wt.% PVPK17/[BMIM][PF6] were effectively applied for RB_19_ dye adsorption.

Based on the above investigation, 1-butyl-3-methylimidazolium hexafluorophosphate [BMIM][PF_6_] was loaded into a polyethersulfone matrix and tested for the persistent and expensive problem of “scaling” in the oil and gas industry. The threshold scale inhibitor was investigated with PES/IL microspheres, where direct interaction with oilfield brine solution occurred without loss of the active ingredient (IL). Performance was assessed in a high-temperature brine solution using chemical screening tests, the dynamic tube block method and electrochemical techniques. The scale inhibition efficiency (82%) was achieved using 25 wt.% of [BMIM][PF_6_] microsphere on the lab scale. IL solubility in water and morphology altered by the IL brings the required novelty. This novel approach led to the prevention of ionic liquid loss during experimentation, thereby improving the deposition of calcium crystals on the solid surface which facilitates their easy removal from the microsphere surface. These fabricated microspheres had longer residual effects of chemical treatment (Figure 12) and were utilized at a cheaper rate due to more effective control and less chemical consumption [57].

Polyethersulfone (PES) microspheres loaded with amine-functionalized imidazolium cation ILs with a common anion bistrifluoromethylsulfonylamide ([TF2N]^−^) were evaluated for adsorption of CO_2_ and CH_4_. Amine-based ILs loaded in the PES matrix exhibited a very limited adsorption capacity of these gas molecules indicating structural defects. A major analysis and interpretation of the defects shows that they may be due to the loss of porosity or unsuitable pore size due to the immobilization of ILs which have large molecular volumes. Hence, it was concluded that an alternative low molecular volume and high-performing IL, as well as different IL loading methods, need to be adapted to improve the performance of IL-PES microspheres [58]. 

## 5. Bio/Green Pore Formers

### 5.1. Deep Eutectic Solvents (DESs) as Porogens in Asymmetric Polymer Membranes

Today, deep eutectic solvents (DESs) are recognized as the new generation of solvents to be tested in different applications and approaches [59,60,61,62]. DESs fall into the guidelines of the “Twelve Principles of Green Chemistry” which led to an exponential increase in their usage. They also offset some of the primary drawbacks of typical solvents and ionic liquids (ILs). In the framework of membrane engineering, DESs have been potentially considered for tailoring membrane structures while aiding in diverse membrane fabrication protocols. Promisingly, in a polymeric membrane containing DESs, both the components (i.e., the polymer and DES) may either interact or not, while still playing an important role in the membrane formation. For instance, DESs could work exclusively as a typical solvent by dissolving the polymer phase [63], or more interestingly, the DES compounds’ hydrogen bond acceptors (HBA) or hydrogen bond donors (HBD) could be involved within the polymerization process [64].

The incorporation of DESs could also result in important changes in polymer properties [65], including surface modifications, as well as morphological and structural changes [66]. There is an increasing demand for highly selective membrane technologies (e.g., pervaporation, gas separation) that have membrane surfaces with a hydrophilic or hydrophobic nature to split polar and non-polar molecules [67,68]. Thus, alteration of the physico-chemical properties and the nature of DESs (i.e., hydrophilicity or hydrophobicity) becomes significant when combined in polymeric membranes. Very recently, DES application in polymer materials has also included the synthesis and different mechanisms for polymer formation, hydrogel design, molecularly imprinted polymers and porous-structured monoliths and membranes [63]. The applications of DESs in various processes in membrane technology is shown in Figure 13.

Towards the preparation of a polymer membrane, the solvent is considered the crucial element for dispersing the polymer network, and is required to create the doped solution and subsequently result in a membrane [70]. In the case of porous polymer membranes, phase inversion is likely to be the most applicable technique, presenting multiple modifications and strategies according to need, such as solvent-induced, temperature-induced or vapor-induced phase separation [71,72]. Typically, when a casting solution is immersed into a non-solvent coagulation bath (generally water), the interchange of solvent and non-solvent takes place due to diffusion, resulting in a phase transition and thus the formation of a membrane [73]. Thanks to these water-soluble properties, some DESs have been proposed as pore-forming candidates.

In recent investigations, Jiang et al. [74] reported the fabrication of polyether sulfone (PES) UF membranes by introducing various imidazole-based DESs, based on organic Cl and Br salts and organic imidazole molecules (IM). Experimentally, the porogen fostered the porosity and enhanced the membrane pore size, which allowed the creation of more permeable membranes. The DESs tend to possess polar or hydrophilic groups [75]. Based on the affinity of the selected DES to water, the exchange rate between the conventional solvent (e.g., NMP) and non-solvent (water) in a coagulation bath could relatively speed up to promote the fabrication of membranes with enhanced porous structures [74,76]. As can be seen in Figure 14, Jiang et al. [75] evidenced the merging of DESs, particularly tetrabutylphosphonium bromide- imidazole (P4444Br/IM), into the PES matrix, which could turn the pores into macro-voids in the resulting structure.

Such a phenomenon was more convincing at the highest DES concentration of ca. 2 wt.%. Regarding the filtration performance, the membranes showed a water permeability as high as 781 L m^−2^ h^−1^, which was approximately six times higher than the pristine PES membrane. Additionally, it was observed that the rejection towards bovine serum albumin (BSA) (Figure 15). remained unchanged by DES usage since it was seen that all the membranes displayed a rejection of over 97%. The authors stated that the high BSA rejection was a result of the narrowly distributed pore diameters as well as the reduction in the effective pore size. During the filtration of humic acid solutions, the water flux decline was less pronounced, with values remaining stable. This shows that DES-based polymer membranes can benefit from the DES enhanced antifouling properties of lower surface roughness. These outcomes agreed with the studies documenting composite polyamide membranes treated with various DESs using choline chloride (as HBA) and ethylene glycol, urea and glycerol (as HBD) [77]. This study declared that the chemical surface modification substantially improved the permeation by 2–5-fold in comparison with the non-treated membrane with unchanged rejection. Such a flux enhancement was associated with the enhanced surface wettability acquired thanks to the DES application, as documented by Jiang and co-workers [75], who found an enhanced surface smoothness for the membranes treated with DESs. Here, the phenomenon was ascribed to the presence of hydrogen bonding between the DES and the polyamide moiety, later verified by zeta-potential analysis [78].

In a more recent work, Vatanpour et al. [79] documented a similar development by preparing DESs using choline chloride and ethylene glycol, forming the well-known ethaline. The resulting hydrophilic DES was subsequently applied as a porogen during the fabrication of PES/polyvinyl pyrrolidone nanofiltration (NF) membranes. The authors gave a concluding remark saying that the application of ethaline contributed to better membranes in terms of (i) uniform pores on the membrane surface, (ii) presence of large macro-voids in the finger-like layer, (iii) an improved surface smoothness, (iv) better hydrophilicity and (v) improved separation performance in both protein rejection and dye removal and permeability.

### 5.2. DESs in the Fabrication of Porous Composite Membranes

Kuttiani Ali [80] synthetized choline chloride:ethylene glycol (ChCl-EG) to prepare composite membranes, where the DES was employed for the post-impregnation of silica nanoparticles followed by their embedding into polyimide UF membranes, as can be seen in Figure 16a,b. In general, the DES impregnation onto the nanoparticles did not affect their morphology, but it resulted in a change of inter-spacing of 3.3 nm in the DES-modified silica, while TEM analysis demonstrated the presence of a porous shell based on nanoclusters over the nanoparticles’ surface. It was speculated that these nanoclusters were associated to the DES in the solid state. After optimization, 2 wt.% of DES-tuned nanoparticles was found to be the optimal loading into the polymer membrane, giving the best mechanical properties. When determining with the membranes’ performance in aqueous phenol solution (containing 30 mg L^−1^), the composite UF membranes showed a water permeability of approximately 300 L m^−2^ h^−1^ and a rejection efficiency of 96%. Figure 16c. illustrates the effect of the tuned silica on the performance of the membranes in comparison to the case without particles. Notably, the DES-doped nanoparticles contributed to an excellent phenol uptake on the hydrated surface of the final silica–polyimide membranes; such adsorption was ascribed to the hydrogen bonding and carboxylic moieties of polyimide. Interestingly, hydrogen bonding in silica (thanks to silanol and siloxane groups) and phenol may also play a role in the efficient phenol removal.

In another work, urea and guanidine hydrochloride (at molar ratio 2:1) were used during the exfoliation protocol of silk fibers (diameter: 20–100 nm and length: 0.3–10 µm) [81]. In this research, Tan and co-workers found that the DES satisfactorily performed the exfoliation treatment of the fibers by permeating into the silk fibers, loosening their structure and breaking the hydrogen bonds. The latter finding decreased the strength of the hydrophobic interactions in the silk. The obtained fibers exhibited optimal mechanical properties with the possibility to test them in vacuum filtration. Researchers discovered their ability to eliminate ions, dyes and protein based on the amphiphilic properties of the silk fibers. For example, membranes (of thickness ca. 18µm) exhibited 97% rejection for dyes, such as Rhodamine B, congo red and methylene blue, and acceptable protein adsorption (>96%). On the other hand, the membranes did not perform satisfactorily in the retention of Cu^2+^ions. Afterwards, Tan et al. [82] further confirmed that silk protein nanofibers produced via a DES-aided extraction procedure would be an alternative for new materials with potential application in tissue engineering according to the great cyto-compatibility, flexibility and mechanical stability of the fibers.

One more interesting finding of DESs in membranes regards the potential facilitated water transport when they are embedded in membranes. Seyyed Shahabi et al. [83], for instance, prepared composite thin-film polyamide reverse osmosis membranes modified with 1 wt.% choline chloride–urea, which resulted in a water permeation increase of up to 56 L m^−2^ h^−1^ and a salt rejection of ca. 96.4%. The result was credited to the ability of the DES to tune the membrane’s surface to enhance its surface hydrophilicity and smoothness, which was associated with the presence of hydroxyl (-OH) functional groups. A different research group also analyzed the application of ChCl-EG DES during the chemical functionalization of graphene oxide (GO) NF membranes [84]. These membranes reported water permeability of 124 L m^−2^ h^−1^, represents a 5–7 times increased permeability than the non-treated GO membrane (with 22 L m^−2^ h^−1^), along with a high rejection towards salt and dyes (ca. 99%). To some extent, the DES significantly influenced the structural properties of GO since (i) it modified the d-spacing of GO, (ii) it decreased the lateral size and (iii) it decreased the wettability properties of the final membrane.

## 6. Marine Derived Polymers as Porogens

Ulvan, a green seaweed-derived sulfated polysaccharide was utilized as a morphology-controlled porogen in polysulfone (PS) membrane fabrication. Competitive results were observed using minimal ulvan concentrations in PS membranes with DMF as a solvent. Conventional analytical tools, porosimetry and SEM, confirmed the morphological changes and pore size increase in the PS matrix. Encouraging results showed that the polysaccharide ulvan, extracted from marine resources, may be an alternative to existing synthetic counterparts [85].

An extension of the above study reported the effect of ulvan as an additive (porogen) in the PSf membrane, where it showed a significant influence on the efficiency and morphological properties. Ulvan (0.5–2.0 wt.%) was explored as a morphology-controlled porogen in polysulfone (PS)/Dimethyl formamide (DMF) membrane fabrication. Favorable porosity enhancement was observed in addition to better flux values with (1 wt.%) ulvan concentration. With the increase in ulvan concentration from 0 to 2.5 wt.% and with DMAc as a solvent, the flux increased from 581.8 to 991.3 LMH. Similarly, with the same increase in ulvan concentration and NMP as a solvent, the flux increased from 571.8 to 974.8 LMH. In addition, promising results were obtained with BSA rejection ranging from 78.53% to 72.36% with DMAc solvent and from 81.19% to 75.49% with NMP as solvent, for a period of 12 h. Such encouraging data on morphology and porosity promotes the use of an alternate resource from marine environments in membrane technology (Figure 17 and Figure 18) [86].

Another sulfated seaweed polysaccharide, κ-carrageenan (KCA), was tested as a pore former to develop highly porous polyphenylsulfone (PPSU) membranes. Morphology reveals that the finger-like structure changed to a porous (porosity 48 ± 3% to 78 ± 2%) structure with uniform pores (~2 nm to 1.1 µm) [87]. Similarly, KCA (0.5, 1.0, 2.0 wt.%) was used as a hydrophilic additive to fabricate polyvinylidene fluoride (PVDF) composite membranes. The addition of KCA induced demixing speed and enhanced the hydrophilic nature (Figure 19), which prevented the dye adhesion on the composite membrane PVDF/KCA surface. 

Y.X. Foong et al. [88] applied eco-friendly natural gum Arabic (GA) as a porogen in polysulfone (PSf) membrane fabrication using green solvents. The presence of GA helped to enhance the hydrophilicity and porosity of the membranes. Subsequently, membrane performance was measured through pure water flux and Congo red (CR) rejection capability. In the presence of DMSO as a green solvent, the membrane displayed a comparable membrane performance (Figure 20) to the membrane prepared by using conventional N-methyl-2-pyrrolidone (NMP) as a solvent. GA’s impact was thus confirmed through porosity, pore size and Congo red (CR) rejection capability. Apart from excellent dye rejection, these membranes also exhibit a flux recovery ratio (FRR) of 93.29% with DMSO as a solvent and additive GA.

## 7. Conclusions

This review summarizes the types of pore-forming materials used to develop pores in various substrates and discusses them in the context of their practical applications based on their porous morphology. The role of these porogens, especially their compatibility with the host structure, is important for the wide scope and performance of various membrane technologies. This study investigates the feasibility and compatibility of fabricating membrane materials with inorganic, organic and green pore forming agents with conventional and green solvents. The overall performance of the membranes with an increase in the concentration of porogens was presented in detail. It was clearly revealed, that in each attempt to fabricate with different porogens, highly reproducible membranes were produced with comparable or better characteristics in terms of pore size, porosity, surface wettability, pure water permeability, flux and rejection parameters. Generally, in all modified membranes, the role of porogens is to increase the formation of pores. Therefore, the increase in the hydrophilicity of the membrane was also thoroughly demonstrated. More particularly, the bio/green porogens exhibited a wonderful performance with very little concentration (0–2 wt.%), with an increase in hydrophilicity in addition to enhanced performance of water flux, rejection and flux recovery without compromising the original membrane performance.

The current challenge is in porogen selection, which is determined on the basis of previous experience, published works or latest investigations. The preliminary experiments were based upon known physico-chemical properties such as physical state, solubility and miscibility and chemical constituents. The next important influential factor is the porogen: co-porogen and/or porogen ratio; the monomer–solvent ratio has to be typically optimized to meet the intended application, which may require specific membrane stability, permeability and/or efficiency. The number of different porogens in membrane fabrication can be one, two, three, or more. The most important property is the nature of the porogens; a greater trend in the shift from toxic and expensive chemicals to low cost, natural, greener chemicals is needed. Hence, it can be assumed that the proper selection of porogens plays an extremely important role in the fabrication of membranes and the improvement in performance. With this observation, we could say that, to date, only relatively few porogens have been used. However, several attempts have been made to develop a standard protocol for porogen selection and application. Currently, extensive research to identify the wide range of available porogens is required, and much more effort is needed to be exerted to make the optimization procedures easier and faster. This novel methodology in this research area will certainly lead to an effective membrane system to expand in the future with a low cost, high safety, ease of removal and possibility to control the porosity and surface area for an innumerable number of applications.

## 8. Outlook

Tailoring the structural and chemical properties of porous materials such as membranes, including pore size, shape, surface roughness, hydrophilicity and connectivity, is a key step in furthering their application within separation technology. The selection of an appropriate porogen is considered an essential criterion in the preparation of membranes. Porogens, at the same time as other factors such as polymerization temperature, time, type and crosslinking ratio, also affect the resultant membrane morphology, surface area, pore volume, permeability and mechanical stability. The field of separation science is at an exciting stage in its evolution. Compared to 20 years ago, there are many more types of membranes which are commercially established and are still developing rapidly. In the recent past, Metal–Organic Frameworks (MOF) membranes and Covalent Organic Frameworks (COF) are gaining attraction and have been widely used in gas separation and liquid separation. They are novel materials with rigid, highly ordered and tunable structures and can actively manipulate the selectivity, holding great potential as next-generation membrane materials for ion separations. However, the application and development of MOF membranes in other fields still has limitations. Even though we have presented many porogen-incorporated membranes and their performances in this review, we suggest that the long-term solution is to develop computational structure/property prediction tools to augment sustainable approaches to apply the novel bio-green porogens to a larger extent.

## Figures and Tables

**Figure 1 polymers-14-05209-f001:**
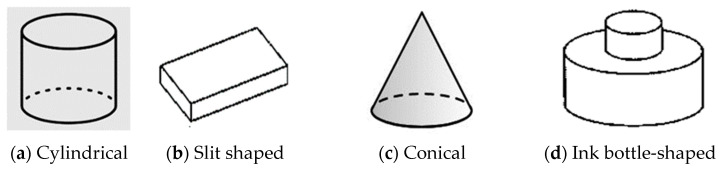
Classification of pores based on geometry.

**Figure 2 polymers-14-05209-f002:**
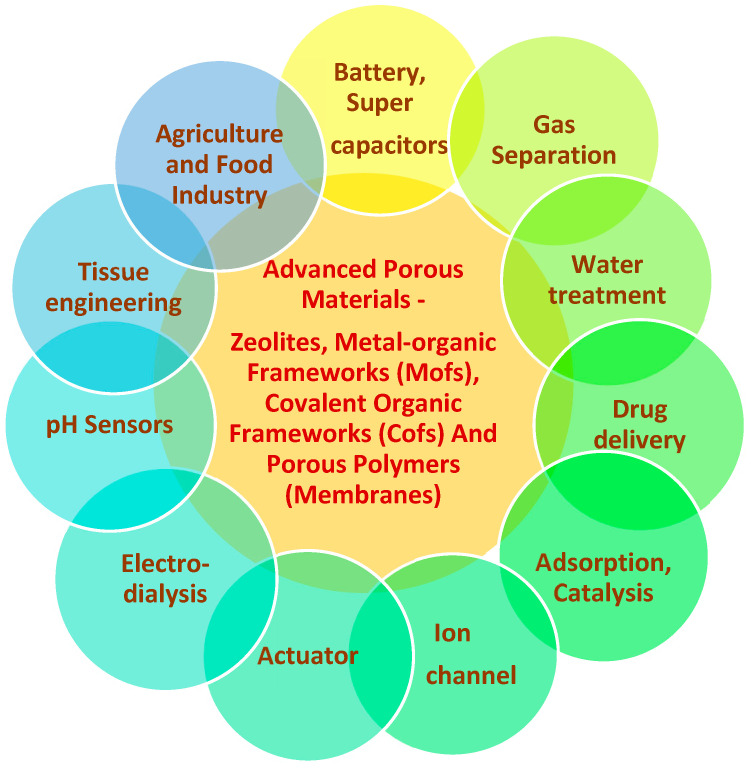
Advanced porous materials in various applications.

**Figure 3 polymers-14-05209-f003:**
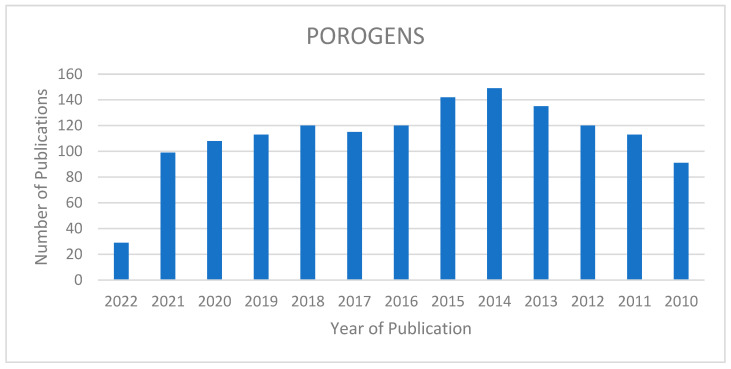
Publication record of material fabrication using different types of porogens over the last decade (https://pubmed.ncbi.nlm.nih.gov, accessed on 22 March 2022.).

**Figure 4 polymers-14-05209-f004:**
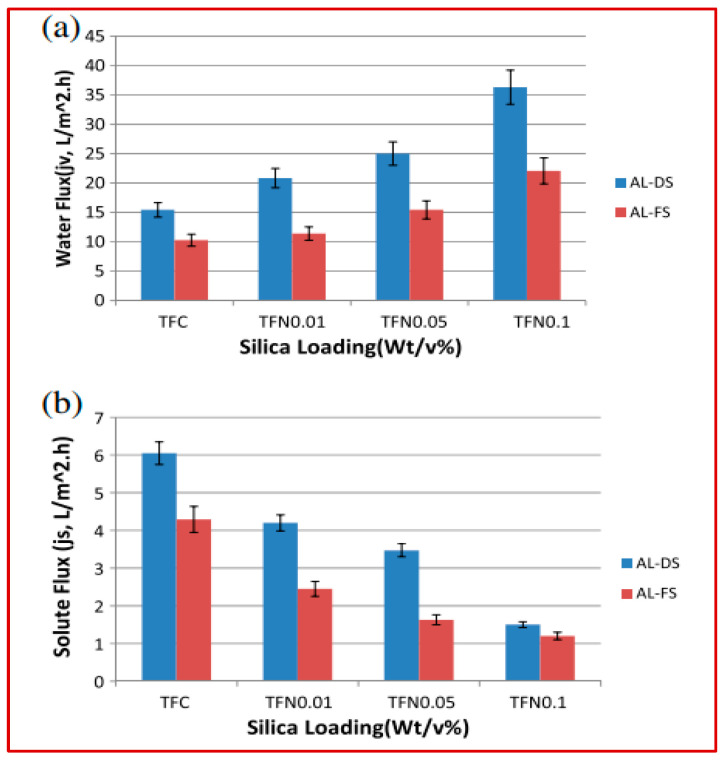
FO water flux (**a**) and solute flux (**b**) in synthesized FO membranes [9].

**Figure 5 polymers-14-05209-f005:**
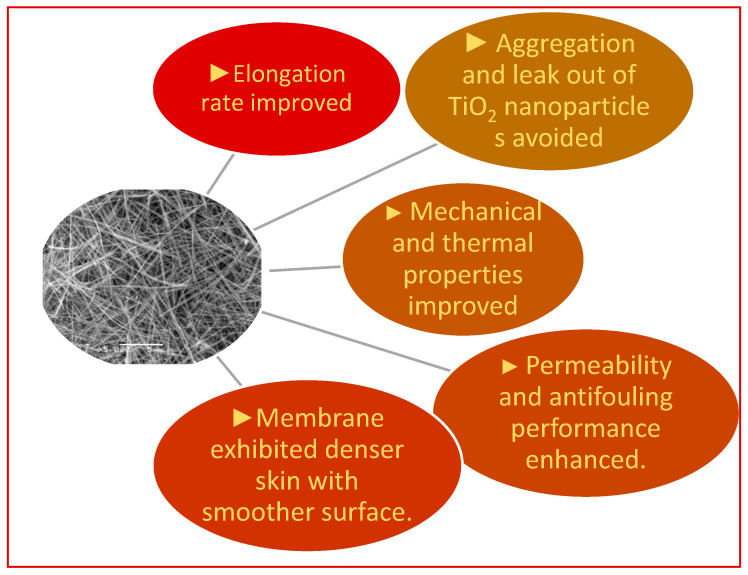
Effect of nanowire incorporation on membrane properties.

**Figure 6 polymers-14-05209-f006:**
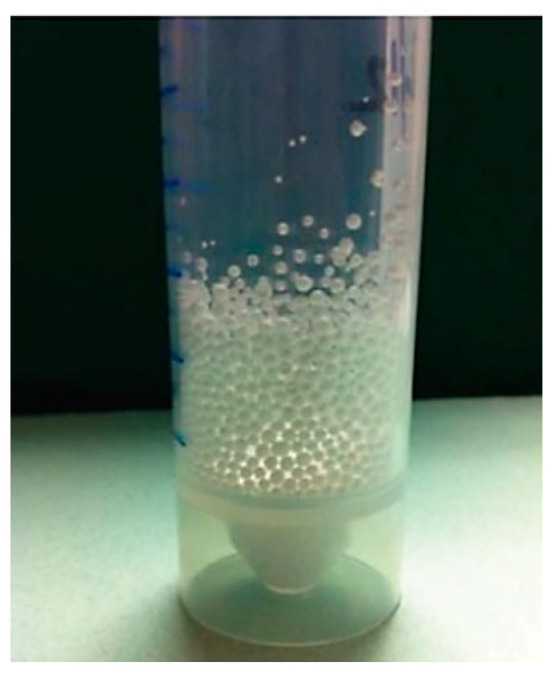
Optical image of the MCs produced [54].

**Figure 7 polymers-14-05209-f007:**
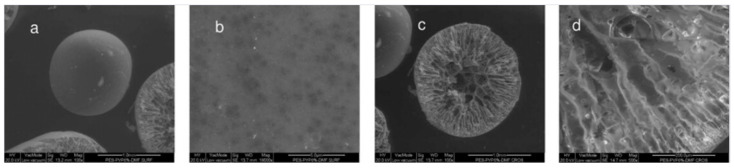
SEM microstructures of 10 wt.% PES/DMF/6 wt.% PVP K17 MCs prepared by using the phase inversion technique: (**a**) surface (100×), (**b**) surface magnification (20,000×), (**c**) cross-section (100×), and (**d**) magnified cross-section (500×) [54].

**Figure 8 polymers-14-05209-f008:**
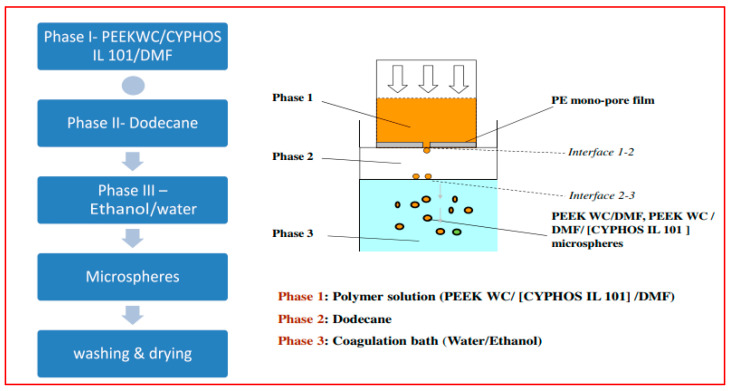
Process flow chart for the encapsulation of CYPHOS IL 101 into PEEKWC microspheres [55].

**Figure 9 polymers-14-05209-f009:**
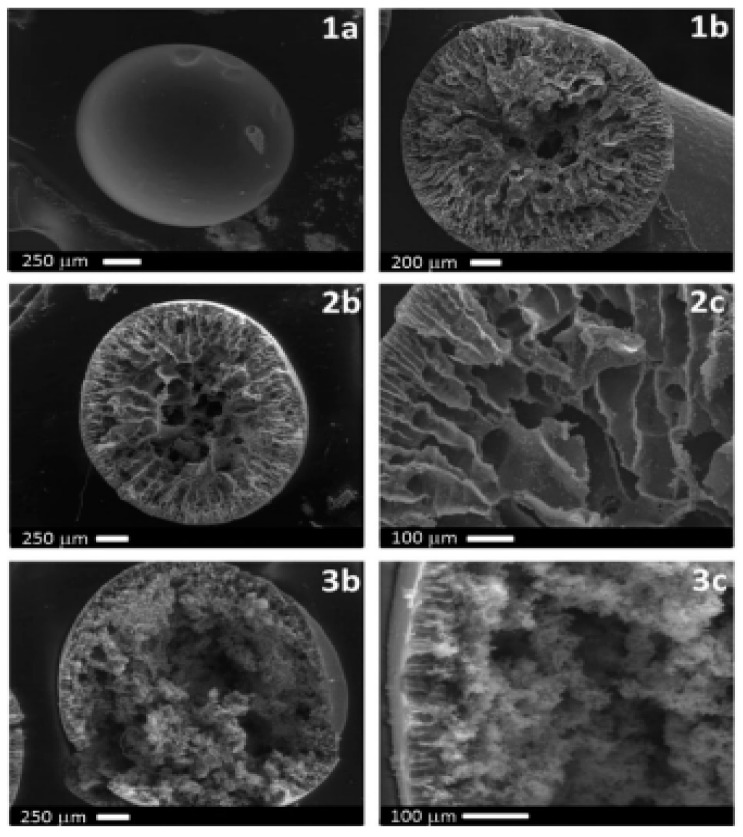
SEM microimages of PES microcapsules prepared by the phase inversion technique with different concentrations of [BMIM][PF6]: (**1a**) External, (**1b**, **2b** and **3b**) Cross-section, (**2c** and **3c**) Magnified cross-section (1-PES Blank, 2-PES/5 wt.% [BMIM][PF6], 3-PES/20 wt.% [BMIM][PF_6_]) [55].

**Figure 10 polymers-14-05209-f010:**
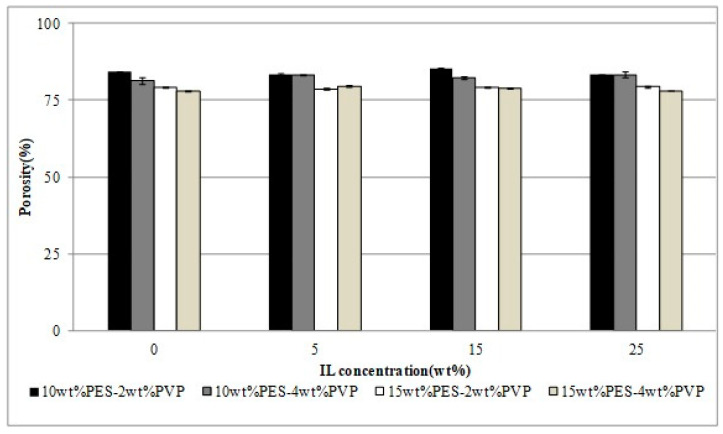
Porosity of PES membrane as a function of PVPK17 concentration [56].

**Figure 11 polymers-14-05209-f011:**
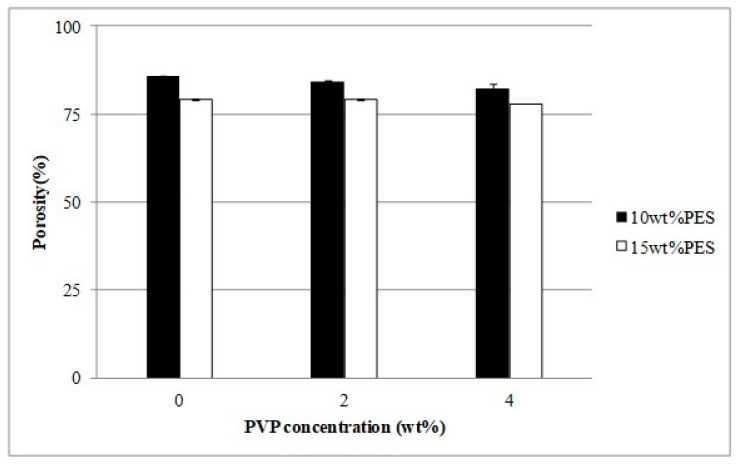
Porosity of PES membrane as a function of IL concentration [56].

**Figure 12 polymers-14-05209-f012:**
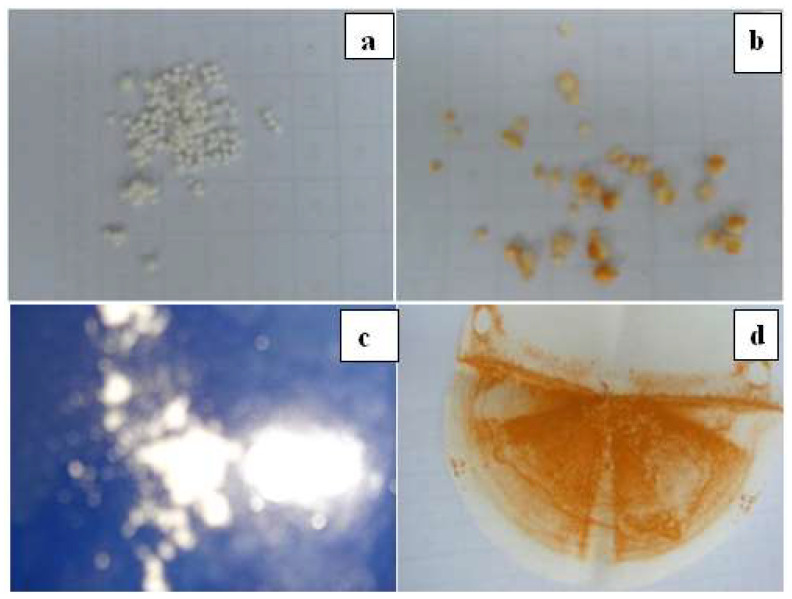
(I) Optical image of PES/[BMIM][PF_6_] microsphere (**a**) before scale treatment and (**b**) after scale treatment. (II) Deposited calcium carbonate scale (**c**) before scale treatment and (**d**) after scale treatment [57].

**Figure 13 polymers-14-05209-f013:**
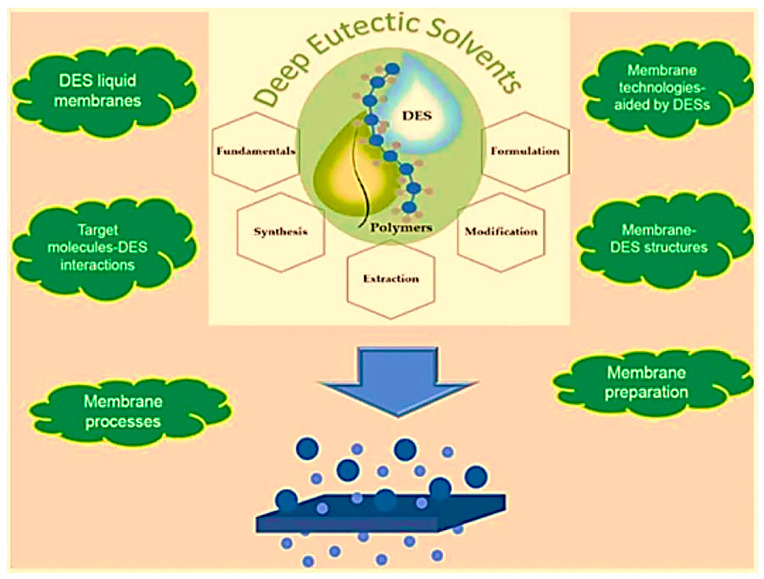
Applications of DESs in various processes in membrane technology [69].

**Figure 14 polymers-14-05209-f014:**
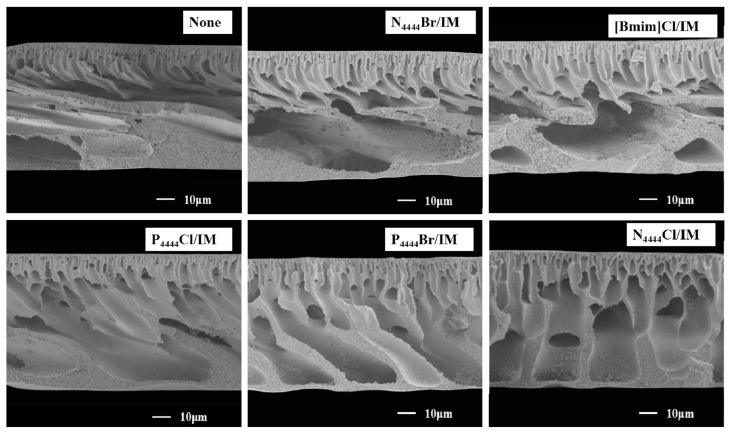
Cross-sectional SEM images of PES membranes with different IM-based DESs as additives [75].

**Figure 15 polymers-14-05209-f015:**
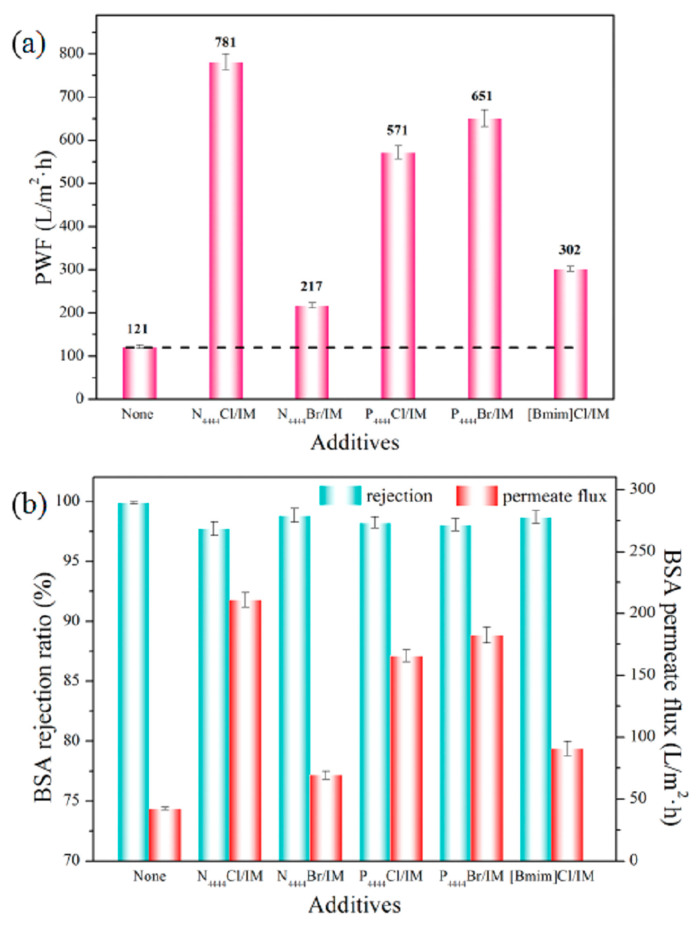
(**a**) Pure water flux and (**b**) BSA rejection and permeate flux of PES membranes with different IM-based DESs as additives. Transmembrane pressure: 2 bar [75].

**Figure 16 polymers-14-05209-f016:**
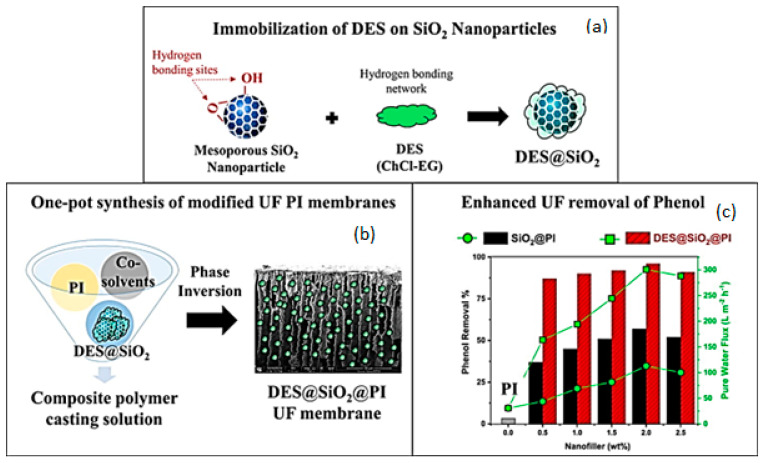
(**a**) Doping of choline chloride–ethylene glycol (ChCl-EG) onto silica (SiO_2_) particles, (**b**) incorporation of DES-doped silica nanoparticles in polyimide UF membranes and (**c**) their performance in filtration of phenol solutions [80].

**Figure 17 polymers-14-05209-f017:**
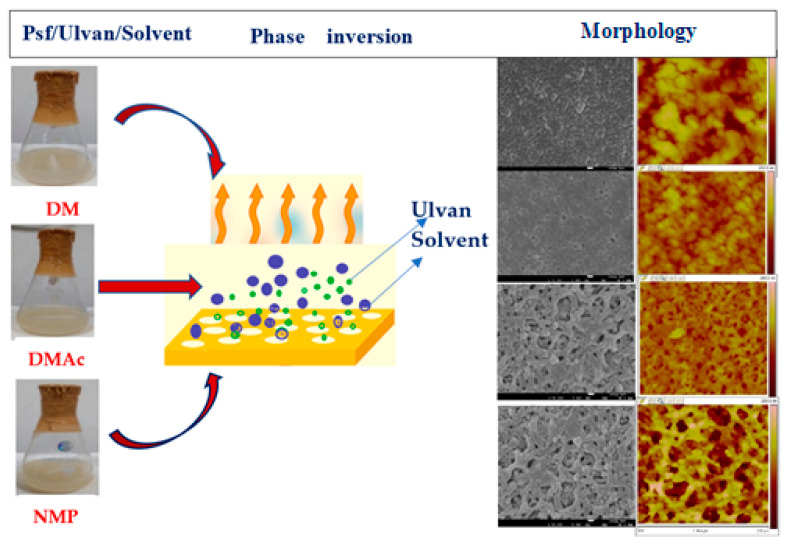
PSF/ ulvan /DMF membrane fabrication and morphology [86].

**Figure 18 polymers-14-05209-f018:**
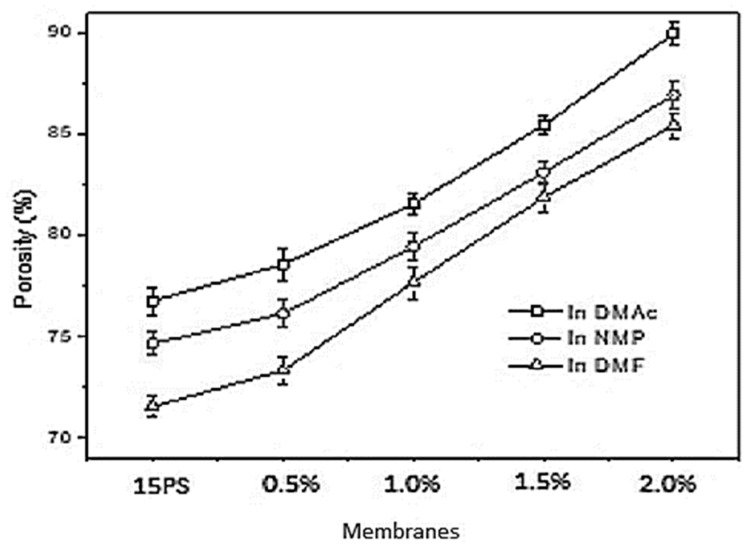
Porosity (%) of Polysulfone–ulvan membranes with different solvents [86].

**Figure 19 polymers-14-05209-f019:**
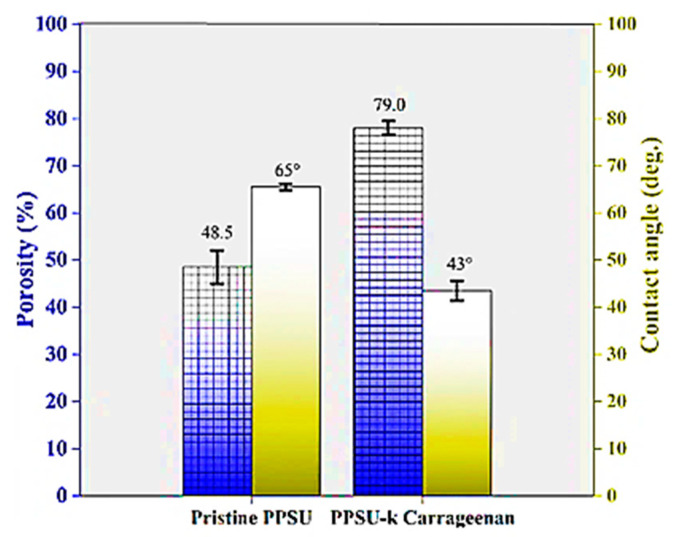
Porosity and contact angle measurements of PPSU and PPSU/kCA membranes [87].

**Figure 20 polymers-14-05209-f020:**
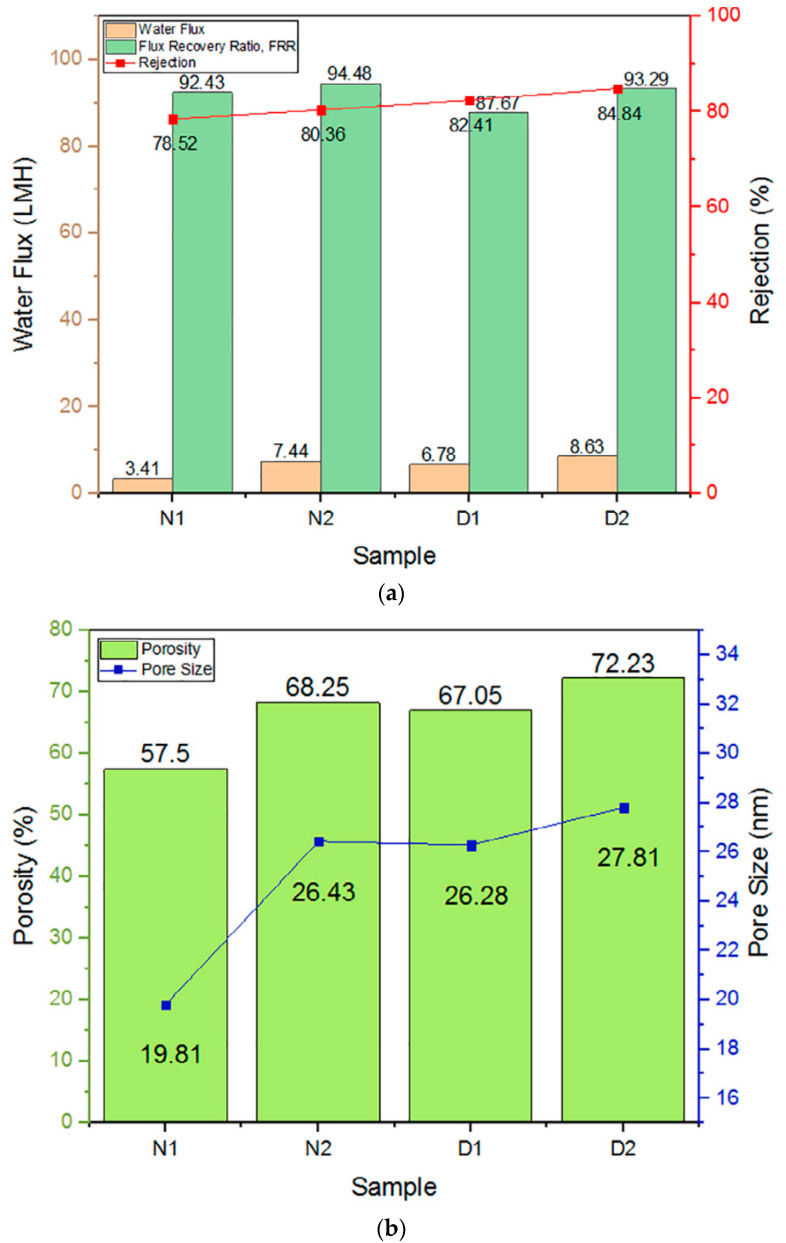
(**a**) Pure Water Flux, Rejection and Flux Recovery Ratio of Membrane. (**b**) Porosity and Pore Size of Membrane [88].

**Table 1 polymers-14-05209-t001:** Various types of porogens with their functions and applications.

S. No.	Porogens Used	Functions	Applications
1.	Water-soluble polymers such as Polyethylene glycol (PEG), poly(vinylpyrrolidone) (PVP), Polyvinyl alcohol (PVA), etc.	Increases mean pore size distribution, membrane porosity and hydrophilicity.	Water filtration. Pharmaceutical and biomedical applications, fuel cell applications, etc.
2.	Low molecular weight inorganic salts: Lithium chloride, Zinc chloride, SiO_2_, etc.	Good adsorption capabilities, high transparency, easy regeneration	Enhanced heterogeneous photocatalysts and their application to various reactions for organic pollutant removal from air and water
3.	Calcium Carbonate	Improved the apparent porosity and enlarged the pore size with good mechanical strength	High apparent porosity ceramics
4.	Ammonium bicarbonate	Enhanced porosity and show more uniform pore distribution Well interconnected macroporous scaffolds were produced having mean pore diameters of around 300–400 μm	Shape memory alloys Highly open porous biodegradable scaffolds for tissue regeneration [8]
5.	Waste biological resource such as saw dust, potato, wheat, corn, rice starch, etc. Corncob bio char from K_2_FeO_4_ and KOH	Reduction composition of pore former results in reduction in membrane pore size and porosity. Porosity also depends on thermal conductivity Good capacitance equivalent to graphene materials	Microfiltration and microbial filtration applications. Burners, anodes, thermal barrier coatings and insulating layers. Capacitive deionization plate materials or electrodes
6.	Fire clay bricks—wastes from renewable or mineral resources	Porosity, water absorption, density, mechanical resistance and even thermal insulation is enhanced and modified.	Innovative building materials
7.	Carbon black Activated carbon from the palm oil shell	Pores were helpful for enhancing the strength and decreasing the thermal conductivity Pore formers increase the porosity and pore volume Shown best membrane permeability	Alumina porous ceramics Proton exchange membrane fuel cells Peat water microfiltration
8.	Naphthalene, carbon beads or fibers polymers such as PMMA (polymethylmethaacrylate), polyurethane, cellulose and paraffin oil	Porosities up to 90% with pores ranging from 1 to 2000 mm in size are reported	Drug delivery and manufacturing of 3D scaffolds with desired porosity
9.	Biodegradable polymer: Chitosan	Produced anodes with lower fracture strength and modest electrical conductivity	Solid oxide fuel cells
10.	Marine source seaweed polysaccharides: κ- Carrageenan	Continuous porous structure with uniformly distributed pores was obtained. It also increased the membrane porosity and mean pore diameter increased	Membranes for various applications
11.	Marine source seaweed polysaccharides: Alginate	Showed a high porosity and an open porous structure	3D Porous hydrogel as meniscus substitute
12.	Marine source seaweed polysaccharides: Ulvan	Very high influence on the efficiency and morphological properties	Ultrafiltration membranes

**Table 2 polymers-14-05209-t002:** Different inorganic nanoparticles used in thin film nanocomposite membranes.

S. No.	Nature of the Membranes	Inorganic Porogens Used	Properties Tuned	Applications
1.	Thin-film nanocomposite (TFN) membranes	SiO_2_ nanoparticles	Higher water permeability, high water flux and better salt rejection	Forward osmosis [9]
2.		Amine functionalized multi-walled carbon nanotubes (F-MWCNTs)		Forward osmosis [10]
3.		TiO_2_ nanoparticles	Higher water permeability and low reverse solute flux	Forward osmosis [11]
4.		Porous zeolite nanoparticles	Higher water permeability and high water flux	Forward osmosis [12]
5.		NaY zeolite nanoparticles		Forward osmosis [13]
6.	Metal matrix membranes	Silica gel	Higher water permeability, high water flux and better salt rejection	Forward osmosis [14]

S. No.

**Table 3 polymers-14-05209-t003:** Different inorganic and mixed inorganic/organic materials incorporated as porogens.

Inorganic Materials	Membrane Type	Polymer	PWF (L/m^2^ h bar)	Refs.
Iron–Nickel oxide	NF	PES	2.20	[23]
Metformin/GO/Fe_3_O_4_	NF	PES	9.02	[24]
Chitosan–Montmorillonate	Loose NF	PES	15.60	[25]
CNT	NF	PES	10.66	[26]
Sulfonated halloysite nanotube	Loose NF	PES	17.00	[27]
SiO_2_/LiCl	UF	CA	18.06	[28]
SiO_2_/PEG600 (inorganic/organic)	UF	CA	23.48	[28]
LiF	UF	PES	100.00	[21]
LiCl	UF	PES	82.00	[21]
LiBr	UF	PES	43.00	[21]
Clay/LiCl	UF	PSf	263.00	[29]
LiCl	RO	Aromatic Polyamide Membranes	34.00	[22]
LiClO_4_	RO	Aromatic Polyamide Membranes	42.00	[22]
ZnCl_2_	RO	Aromatic Polyamide Membranes	19.00	[22]
Mg(ClO_4_)_2_	RO	Aromatic Polyamide Membranes	38.00	[22]
ZnCl_2_+ Pyridine Hydrochloride (inorganic/organic salt)	RO	Aromatic Polyamide Membranes	46.20	[22]
Mg(ClO_4_)_2_ + Pyridine Hydrochloride (inorganic/organic salt)	RO	Aromatic Polyamide Membranes	189.00	[22]

(NF—Nanofiltration; UF—Ultrafiltration; RO—Reverse Osmosis; PES—Polyethersulfone; CA—Cellulose acetate; Psf—Polysulfone).

**Table 4 polymers-14-05209-t004:** Different water-soluble porogens used in ultrafiltration (UF) membranes.

S. No.	Water Soluble Porogens	Structure	Polymers and Membranes	Properties Enhanced
1	PEG	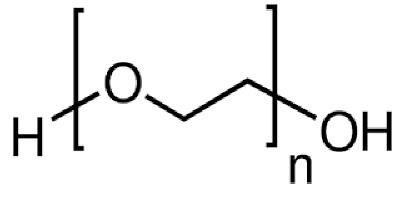	Modified cellulose acetate UF membrane	Porosity/permeability of membrane increased [30]
			Asymmetric CA membranes	Increased pure water flux and macro-void formation [31]
			PEI membrane	Altered membrane morphology and larger average pore radius [32]
			PVC UF membranes	High water flux, excellent thermal stability and mechanical strength
2	PVP	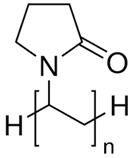	PVDF hollow fiber membrane	Effective porosity, consistent mean pore size [33,34,35]
			PEI hollow fiber membrane	Larger pore size [36,37]
			PES hollow fiber membrane	Enhancement of water flux [38]
			PES UF membrane	Membrane permeability [39]
3	PVA	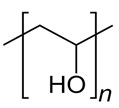	Alumina	Enhanced porosity [40]
			Adsorptive polymer chitosan conjugate	Heavy water removal [41]
			Ethyl cellulose film coated pellets	Increased drug dipyridamole release [42]
			PVA membranes	Separation of CO_2_ from water [43]
4	PAM	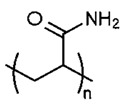	Ceramic membranes	Alters the membrane pore size and membrane flux [44]
			Chitosan–PAM membrane	Thermal stability of the membrane enhanced [45]
			Poly(ether ether ketone) (PEEK-WC) membrane	Decreased hydrophobicity
			PAM–polydivinylbenzene membranes	Enhanced pH stability and separation factor
5	PAA	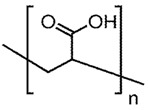	PI membranes	Enhancement of porosity [46]
			PSf membrane	Increased rejection of lead and sulphur dyes [47]
			Nylon 6, six composite membranes	Improved flux rate and separation factor [48]
			Microporous membrane with polypropylene	Chemical valve effect on flux by varying pH [49]
6	HPMA	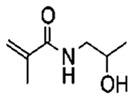	HPMA-PSf membrane	Effective boron removal [50]
			PS-b-HPMA membrane	Enhanced pore size and water flux [51]
			Bisphosphonate-derived ligand membrane	Improved flexibility of the copolymer ligand and superior specific protein adsorption [52]
			Amphiphilic p(HPMA)-co-p(LMA) polymeric membrane	Hydrophobic interactions of lipids were prevented and membrane proteins incorporation were allowed [53]

## Data Availability

Data presented in this study are available on request from the corresponding author.

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
