# Peer review of "Emerging Trends in Porogens toward Material Fabrication: Recent Progresses and Challenges"

_polymers, 2022, doi:10.3390/polym14235209_

Round 1
Reviewer 1 Report
The manuscript entitled " Emerging trends in porogens toward material fabrication: Recent Progresses and Challenges " is a review based on 88 references including a patent. It should be emphasized that the cited works come mainly from the last few years, i.e. they cover the latest state of knowledge in the described field. The authors started with an introduction on porosity and types of pores, and then described the porogens used, dividing them into inorganic, organic, bio / green and marine derived polymers.
The article is written correctly. Some minor errors do not diminish the work that in my opinion is almost ready for publication. However, I have a few comments:
The fragments that need improvement or correction:
1. Table 3 shows, not only inorganic, but also mixed porogens: inorganic-polymer [28] and inorganic-organic[22]. The table header should take this into account.
2. Table 4 contain mainly polymeric porogens (in fact only last on is simple organic compound).
3. Although some used abbreviations are quite common, it is good practice to clarify them when first used, so that the reader has no doubts.
Some editorial corrections should be done, as well.
Author Response
Response to reviewer I comments:
Reviewer Comments Point 1: Table 3 shows, not only inorganic, but also mixed porogens: inorganic-polymer [28] and inorganic-organic [22]. The table header should take this into account.
Response: As suggested, the title was modified and we have also indicated the nature of the porogen in brackets.
Reviewer Comments Point 2: Table 4 contain mainly polymeric porogens (in fact only last on is simple organic compound).
Response: As per the suggestions given, we have modified the title as organic/polymeric porogens
Reviewer Comments Point 3: Although some used abbreviations are quite common, it is good practice to clarify them when first used, so that the reader has no doubts.
Response: As suggested, we have expanded the abbreviations used
Reviewer Comments Point 4: Some editorial corrections should be done, as well.
Response: Editorial corrections were carried out.
Reviewer 2 Report
The manuscript entitled "Emerging trends in Porogens toward Material Fabrication: Re- 2cent Progresses and Challenges summarized and discussed the various applications and prospects of g-Porogens in recent years. It can provide some useful information and theoretical reference for the application of Porogens in potential diverse fields. Therefore, the paper can be accepted for publication with major revision based on the following suggestions:
1. The logic of the Introduction is confusing and needs to be re-organized and re-segmented.
2. As a review paper, authors should add more comments and summaries about this field, rather than only listing the related literatures. More comments like the current research situation, the challenge in this filed, as well as analyzing and comparing for the published works should be done
3. In the summary and outlook section, the discussion on challenges and perspectives seems not sufficient. It would be better if more specific direction and approaches could be suggested to address the challenges associated with the porogens in various applications. For example, practicality and commercialization of porogens-based Martials.
4. In the outlook, the authors should further prospect how the computational simulations may aid in the development of porogens-based Martials design and its physicochemical properties. Such detailed remarks should be presented.
Author Response
Response to reviewer II comments:
Reviewer Comments Point 1: The logic of the Introduction is confusing and needs to be re-organized and re-segmented.
Response: As per the reviewer suggestions, Introduction part is reorganised and resegmented.
Reviewer Comments Point 2: As a review paper, authors should add more comments and summaries about this field, rather than only listing the related literatures. More comments like the current research situation, the challenge in this filed, as well as analyzing and comparing for the published works should be done
Response: As per the suggestions, we have included some more summaries and comparative study from other published works.
Reviewer Comments Point 3: In the summary and outlook section, the discussion on challenges and perspectives seems not sufficient. It would be better if more specific direction and approaches could be suggested to address the challenges associated with the porogens in various applications. For example, practicality and commercialization of porogens-based Martials.
Response: As suggested we have included the discussion on challenges and perspectives in conclusion part.
Reviewer Comments Point 4 : In the outlook, the authors should further prospect how the computational simulations may aid in the development of porogens-based Martials design and its physicochemical properties. Such detailed remarks should be presented.
Response: We think this is an excellent suggestion. We will explore further on computational simulations to apply porogens for membrane fabrication in future. As suggested, we have included the outlook section with future prospects of porogens.